# Reprogramming feedback strength in gibberellin biosynthesis highlights conditional regulation by the circadian clock and carbon dioxide

Alexander R. Leydon[1], Leonel Flores[1], Arjun Khakhar[2], Jennifer L. Nemhauser[1]*

1 Department of Biology, University of Washington, Seattle, Washington, United States of America,
2 Department of Biology, Colorado State University, Fort Collins, Colorado, United States of America

* jn7@uw.edu

## Abstract

The phytohormone gibberellin (GA) is an important regulator of plant morphology and reproduction, and the biosynthesis and distribution of GA *in planta* is agriculturally relevant to past and current breeding efforts. Tools like biosensors, extensive molecular genetic resources in reference plants and mathematical models have greatly contributed to current understanding of GA homeostasis; however, these tools are difficult to tune or repurpose for engineering crop plants. Previously, we showed that a GA-regulated Hormone Activated CAS9-based Repressor (GAHACR) functions *in planta*. Here, we use GAHACRs to modulate the strength of feedback on endemic GA regulated genes, and to directly test the importance of transcriptional feedback in GA signaling in the model eudicot *Arabidopsis thaliana.* We first adapted existing mathematical models to predict the impact of targeting a GAHACR to different nodes in the GA biosynthesis pathway, and then implemented a perturbation predicted by the model to lower GA levels. Specifically, we individually targeted either the biosynthetic gene *GA20 oxidase* (*GA20ox*) or the GA receptor GID1, and characterized primary root length, flowering time and the transcriptome of these transgenic lines. Using this approach, we identified a strong connection between GA signaling status and the circadian clock, which can be largely attenuated by elevated carbon dioxide levels. Our results identify a node in the GA signaling pathway that can be engineered to modulate plant size and flowering time. Our results also raise concerns that rising atmospheric $CO_2$ concentration are likely to reverse many of the gains of Green Revolution crops.

## Introduction

Gibberellins (GA) are phytohormones heavily involved in regulating plant growth, influencing many processes including breaking seed dormancy, promoting cell expansion and positively regulating flowering by degrading the repressive DELLA

**Data availability statement:** All code is freely available at Github: https://github.com/achillobator/GAHACR RNA-Seq data has been uploaded to Dryad: https://doi.org/10.5061/dryad.547d7wmgr.

**Funding:** This work was supported by grants to JLN from the National Science Foundation (MCB-1411949) and the National Institutes of Health (R01-GM107084, and R35-GM148135), as well as support from the Howard Hughes Medical Institute Faculty Scholars Program to JLN. ARL was supported as a Simons Foundation Fellow of the Life Sciences Research Foundation. The funders had no role in study design, data collection and analysis, decision to publish, or preparation of the manuscript.

**Competing interests:** The authors have declared that no competing interests exist.

proteins [1–7]. The GA signaling pathway has been modulated in agricultural breeding programs to create widely used cultivars that exhibit semi-dwarf phenotypes by either reducing GA production or perception [8]. High-yield GA mutants in rice and wheat played a pivotal role in the green revolution, which increased agricultural yields globally by as much as 44% [9]. For example, *Semidwarf1* (SD1) in rice (*Oryza sativa*), which encodes a GIBBERELLIN 20 oxidase (GA20ox) [10].

The GA biosynthetic pathway synthesizes GA through a Geranylgeranyl Diphosphate (GGDP) precursor, in a well-defined pathway [11]. Bioactive GAs are synthesized in the final steps of biosynthesis where $GA_{12}$ is converted into bioactive forms of GA ($GA_1$, $GA_3$, $GA_4$, $GA_7$) by GA20ox and GA3 oxidase (GA3ox, Fig 1A). These bioactive forms of GA bind to the GA receptor GIBBERELLIN INSENSITIVE DWARF (GID1) family proteins to form a complex [12,13]. The GA-GID complex binds to the DELLA proteins [14], catalyzing their ubiquitination and proteasomal degradation [15,16], and relieving DELLA-induced repression of GA-regulated genes [17–20]. The GA relief of repression signaling pathway exhibits feedback mechanisms to control the homeostatic levels of key signaling components (Fig 1A, blue lines) [21–23], and has been mathematically modeled by [24].

In addition to feedback regulation by the GA pathway, the GA signaling pathway is regulated by other interconnected signaling pathways. The GA signaling network is interconnected with the phytohormone auxin [18,25], the circadian clock [26], and the network that governs flowering time, just to name a few [7,27,28]. GA signaling is also subject to developmental and environmental states [29,30], and of specific interest is the intersection of atmospheric carbon dioxide ($CO_2$) and GA on development. Elevated carbon dioxide (750 ppm) uncouples GA-required growth control, and rescues exogenous treatment of the GA biosynthesis inhibitor paclobutrazol (PAC) in *Arabidopsis* [31,32], and of GA biosynthesis mutants in tomato [33]. These studies demonstrate that carbon dioxide genetically interacts with the GA signaling network to suppresses the GA-requirement for growth under elevated carbon dioxide levels and this activity may similarly suppress high-yield GA mutant crop cultivars in agricultural settings.

In previous work, we engineered Hormone Activated Cas9-Based Repressor (HACRs), which are synthetic proteins consisting of a deactivated CAS9 (dCAS9) fused to a phytohormone-specific degron and the first 300 amino acids of TOPLESS repressor proteins [34]. These synthetic tools allow hormone-dependent repression of both synthetic and endogenous promoters to report on hormone levels or modulate target gene expression. As the HACRs are repressors, we mathematically modelled the predicted effects of targeting a line positive feedback within the GA signaling network, such as the positive regulation of *GA20ox* genes by DELLAs [24,35,36]. Here, we demonstrate a model-guided intervention in the GA signaling network to tune down GA20 oxidase transcript abundance in a GA-dependent manner, thereby reducing endogenous positive feedback. We observed reductions in root growth, flowering time and *GA20ox* gene expression, as predicted. Similar to previous studies, we observed the ability of elevated carbon dioxide to suppress the low-GA phenotype. Surprisingly, we also observed a subtle modulation of the circadian clock under elevated carbon dioxide, and partial transcriptional rescue of the *GA20ox* feedback intervention. These findings suggest that multiple levels of the GA homeostatic pathway are influenced at elevated carbon dioxide.

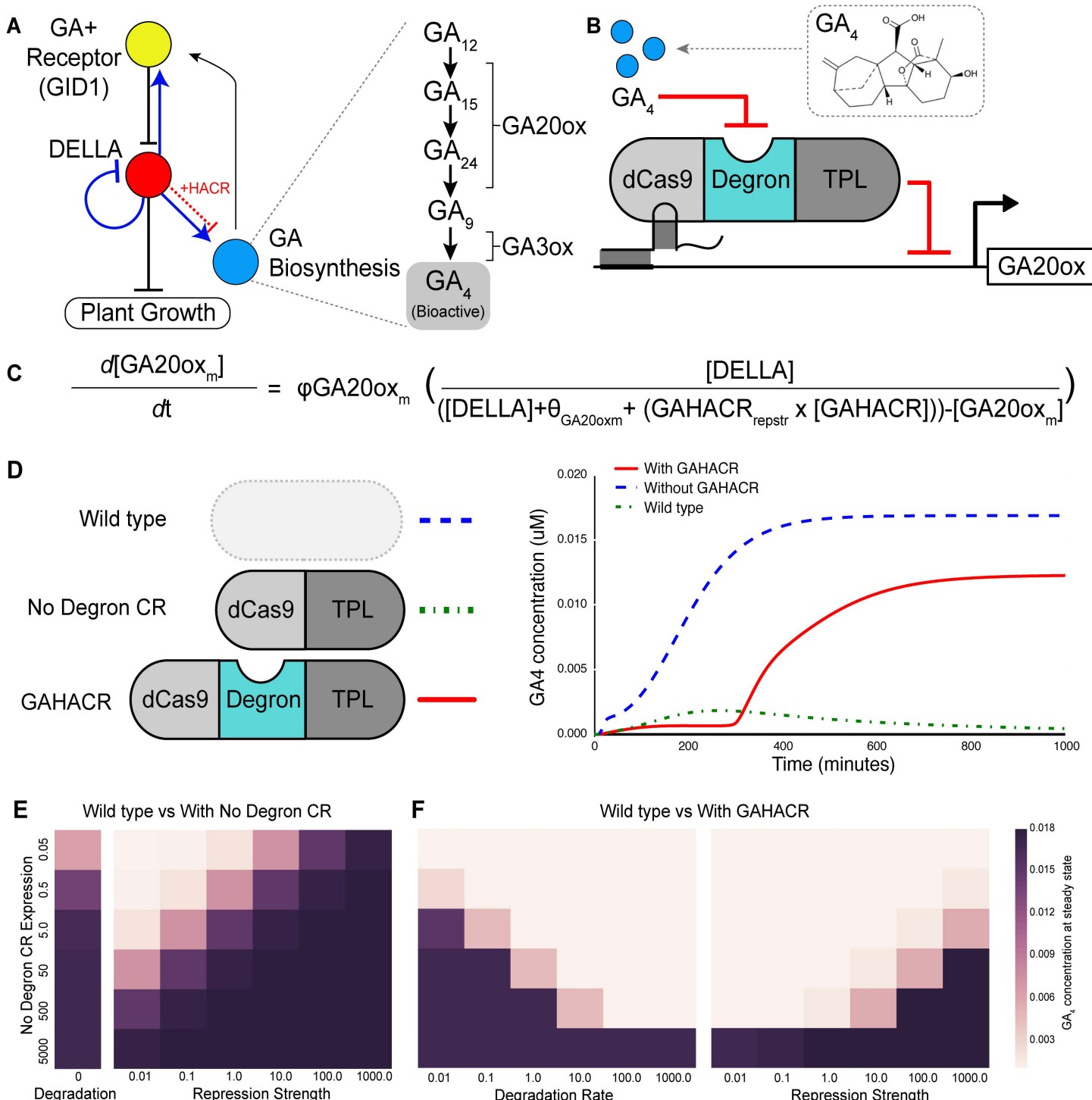

**Fig 1. Modelling the effect of GAHACR integration into the GA signaling pathway.** (A) Simplified network representation of the GA signaling pathway. The GA receptor (GID1 family) + GA node (yellow) negatively regulates the protein levels of the DELLA family repressor node (red). The DELLA node negatively regulates GA target genes and plant growth. The DELLA node also transcriptionally regulates other nodes in the GA pathway to control GA homeostasis (blue lines). The GA biosynthesis node (blue) comprises the multistep pathway that produces bioactive GA such as GA4, expanded in the pathway on the right. A representative line of negative regulation for a GAHACR targeted to GA20ox has been added (dotted red line +HACR). **(B)** A

general schematic of GAHACR protein targeted to the promoter of the *GA20ox* gene see methods for full term breakdown. **(C)** The equation for the influence of GAHACR on *GA20ox* transcriptional output. To simulate the GAHACR's or No Degron CR's repression of GA20ox expression, we incorporated a term that captures GAHACR protein concentration scaled by a user defined repression strength constant (GAHACR$_{repstr}$) into the denominator of the Hill function used to simulate the activation of GA20ox by the DELLA protein. The term GAHACR represents the abundance of the GAHACR protein in the cell. **(D)** Left – a cartoon schematic of the three models compared for different perturbations: wild type – blue dashed line, a No Degron CAS9 repressor (No Degron CR) – green dashed line, and for a GAHACR – red solid line. Right – Predicted dynamics of GA4 concentration for WT, No Degron CR, and GAHACR over time. E-F. GA4 concentration at steady state over a range of values for HACR degradation rate and repression strength for **(E)** No Degron CR, and **(F)** GAHACR.

## Results

### Mathematical modeling can be used to predict the impact of modulating feedback in the GA biosynthesis pathway

The Gibberellic Acid (GA) signaling pathway is a release of repression type hormone signaling pathway, where the receptor complex binds the ligand GA to negatively regulate DELLA repressors and activate GA-responsive transcription ([11,30], Fig 1A - black arrows). The DELLA repressors control GA homeostasis by feedback regulation of the GA pathway through at least three methods: 1) DELLAs activate transcription of the GA biosynthesis genes *GA20 oxidase 2* (*GA20ox2*) and *GA3 oxidase 1* (*GA3ox1*) [23], 2) DELLAs activate the transcription of the receptor *GID1* [12,23], and 3) DELLAs repress their own transcription (Fig 1A - blue lines, [23,24]). We hypothesized that the strength of feedback regulation implemented by the DELLAs on GA perception and biosynthesis are important determinants of the steady state level of GA in a tissue. We further hypothesized that targeting a GAHACR to regulate the genes that mediate perception and biosynthesis (Fig 1A, blue arrows – DELLA to GA20ox, DELLA to GID1) would implement GA dependent repression and thus result in an effective decrease in the GA dependent transcriptional activation of these genes. For example, when targeted to a *GA20ox* promoter (Fig 1A, red dashed line), the GAHACR would repress transcription counteracting the activation through DELLA proteins in a GA-dependent manner (Fig 1B). We hypothesized that this feedback modulation would result in a decrease in the steady state levels of GA *in planta* as well as a change in the dynamics of GA accumulation.

To test our predictions, we simulated this perturbation by first implementing the model of GA signaling proposed by Middleton et al. in python. We then added additional differential equations to simulate GAHACR transcription, translation, and GA-dependent degradation paralleling the approach used for DELLA proteins. However, as the GAHACR is a repressor rather than an activator of the target gene, it's impact on transcription was appropriately altered (Fig 1C, Methods). We also built versions of this model without a GAHACR and without GA triggered degradation of the GAHACR, to simulate a dCAS9 repressor lacking the GA degron (No Degron CR) (Fig 1D). There is qualitative difference in both dynamic and steady state behavior of the GA biosynthesis pathway depending on if a GAHACR or a No Degron CR are targeted to a *GA20ox* locus. We modeled the GA4 concentration at steady state levels across a range of GAHACR expression and repression strengths (Fig 1E, 1F). The decrease in GA4 level when a GAHACR is targeted to *GA20ox* (compared to the no GAHACR control) is directly proportional to the expression level and repressive strength and inversely proportional to the degradation rate of the GAHACR (Fig 1E, 1F). There is a non-linear relationship between expression and repression strength's impact on GA4 levels when a GAHACR targets *GA20ox* as compared to the more linear relationship with a No Degron CR. This highlights the impact of having a feedback loop, and demonstrates that the GAHACR is predicted to be qualitatively different from a No Degron CR.

### GA regulated phenotypes such as plant growth and flowering time can be predictably altered using GAHACRs

The use of the dCAS9 protein as the DNA-targeting domain of the HACR allows for rapid re-targeting of HACRs to genes *in vivo* [34]. This is useful when targeting genes in families with multiple redundant members such as the *GA20ox* gene family (five genes *GA20ox 1–5*, Fig 2A, [37]). We targeted the promoters of the four *GA20ox* genes that demonstrated the highest levels of transcription [11,21]) with sgRNAs assembled onto one T-DNA (GA20ox1–3 & GA20ox5, Fig 2B,

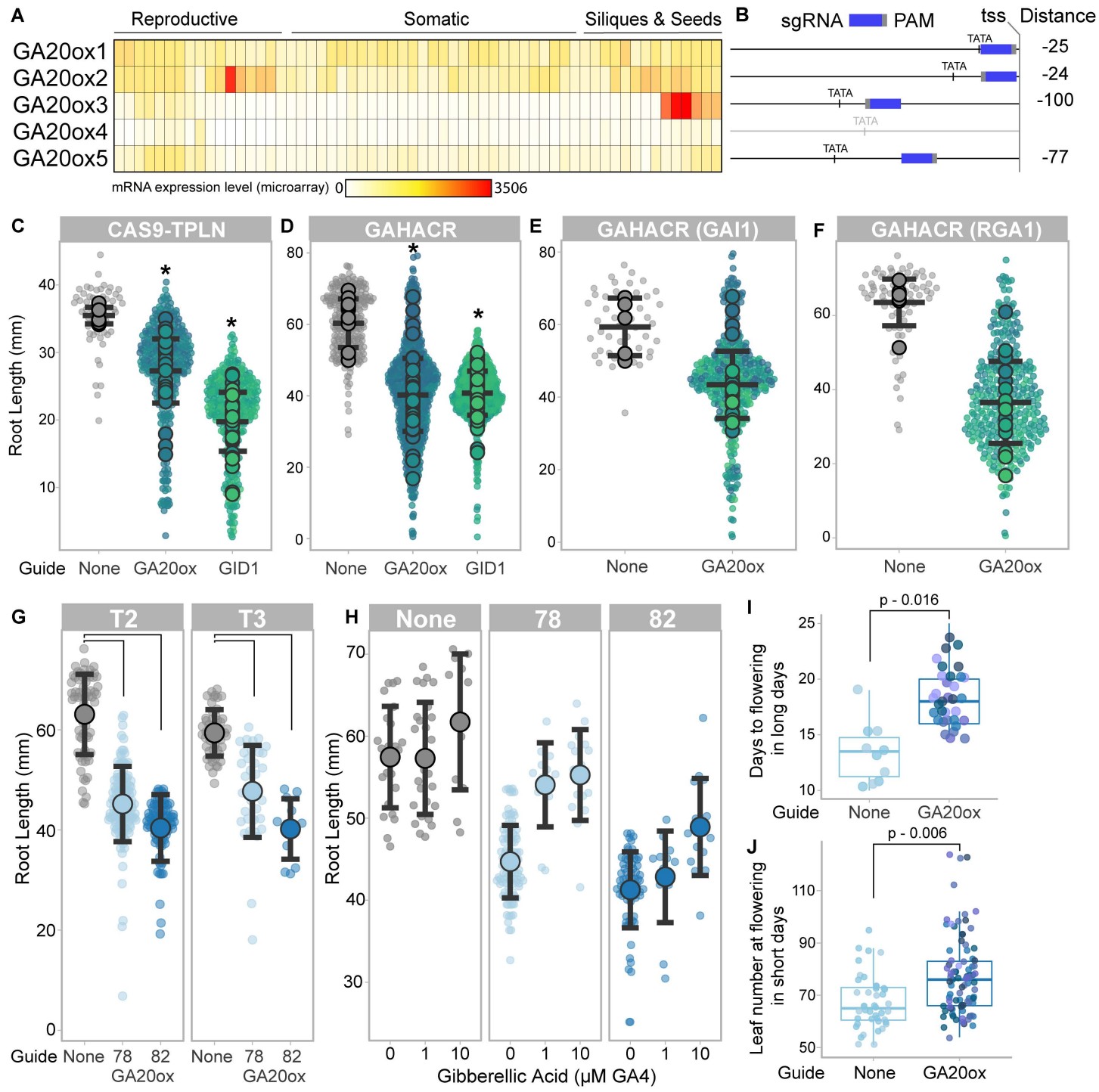

**Fig 2. GAHACR Perturbation of GA of signaling pathways in *Arabidopsis* can modulate root elongation and flowering time. (A)** Expression heatmap of GA20 oxidase family genes (*GA20ox1-5*). All GA20 oxidases are detected at the RNA level, with *GA20ox4* at lowest levels, especially in somatic tissue types. Microarray data, heatmap created with http://pheattreecit.services.brown.edu/. **(B)** Position of the sgRNA in the *GA20ox* promoters: sgRNA in blue, PAM in gray, distance of sgRNA from transcriptional start site (tss) on right. Promoters are aligned by tss. Primary root length in T2 plants is reduced when a CAS9-TPLN repressor **(C)** or a GAHACR **(D)** is targeted to *GA20ox* or *GID1* promoter regions. **(E-F)** GAHACRs targeting *GA20ox* promoters are subset by DELLA degron type, either GAI1 (E) or RGA1 (F). C-F. Each T2 line's mean is shown as a large, filled circle, with corresponding small dots for each individual seedling. Statistics are representative of the mean and standard deviation, and asterisks indicate a p-value < 0.001. **(G)**

Two representative lines GAHACRs targeting *GA20ox* promoters were assayed for root length reductions in both T2 and T3 generations to demonstrate construct stability over multiple generations. **(H)** GAHACRs targeting *GA20ox* promoters show a rescue in root length upon application of exogenous GA (GA4) in a dose-dependent manner. **(I)** Flowering time is delayed in GAHACRs targeting *GA20ox* promoters. Plants were grown in long day conditions and days to flowering was measured starting from the day of plating. **(J)** Flowering time is delayed in GAHACRs targeting *GA20ox* promoters grown under short day conditions. Flowering was measured as leaf number at time of flowering. For (I-J) statistical significance was tested by ANOVA and p-values of comparisons are shown above each graph.

S1A Fig). We introduced these T-DNAs into *Arabidopsis* GAHACR lines containing either a CAS9-TPLN (no degron) or a CAS9-GAdegron-TPLN (GAHACR) and observed significant reductions in primary root length (Fig 2C, 2D). We focused on the GAHACR retargeted to GA20 oxidases for simplicity, and because they allowed us to test the effect of tuning transcriptional feedback, which was our goal. We saw a significant reduction in root length for GAHACRs constructs using either GAI1 or RGA1 as the degrons (Fig 2E, 2F). We chose to further characterize two representative GAHACR lines with the RGA1 degron targeting GA20ox and observed consistent reductions in root length in both second and third generations after transformation, suggesting genetic stability of the components (sgRNA, GAHACR) of this repression scheme (Fig 2G).

We observe the predicted reduction in primary root length in lines that have a GAHACR targeted to *GA20ox* compared to parental controls without the gRNAs, consistent with lowered levels of GA (Fig 2D–2F). We tested the ability of exogenously applied GA to rescue the effect of the GAHACR repression of *GA20ox* and observed a rescue of root elongation (Fig 2H). This rescue by exogenous GA supports the effectiveness of the GAHACR as a conditional repressor. We sought additional support that *GA20ox* repression in adult plants resulted in GA-deficient phenotypes, and so we measured the length of time required for flowering as GA plays a positive role in this developmental transition. We observed a delay in flowering time in lines that have a GAHACR targeted to *GA20ox* compared to parental controls without the gRNAs in long day conditions (Fig 2I). As the contribution of GA to flowering time in long day conditions is relatively minimal compared to other inputs [7], we evaluated the effect of our perturbation in plants grown in short day conditions where the role of GA is much more pronounced. We observed a more significant delay in flowering times under short day conditions (Fig 2J), corroborating the previous observations under long day conditions.

The mathematical model predicted that repression of *GA20ox* genes would reduce the cellular concentration of bioactive GA (Fig 1). Phenotypic analysis of the lines with GAHACRs targeting *GA20ox* promoters demonstrated three lines of evidence that GA20ox activity had been reduced: (1) reduced root elongation (Fig 2C–2F) that is (2) rescued by exogenous GA application (Fig 2H), and (3) an extension in time before flowering (Fig 2I, 2J). In order to map the transcriptional landscape that is modulated by tuning down feedback between DELLAs and *GA20 oxidases*, we performed RNA-seq to quantify changes in the transcriptome. Seedlings were grown under long day conditions and RNA-seq was performed on 10-day old seedlings at ZT8 (Fig 3A). We observed a small number of differentially regulated genes (DEGs, 172 genes) that demonstrated clear overlap in behavior between the two biological replicates (Fig 3B, S2 Fig, S1 Table). When we examined this list of DEGs and intersected them against genes with altered expression in GA mutants, GA treatments, and Paclobutrazol treatments [38–40]. We observed a partial overlap with known GA-related genes (82/172 genes, Fig 3C). When we examined the upregulated DEGs by Gene Ontology (GO) analysis we observed enriched GO terms related to photosynthesis, light perception or response, and circadian rhythms (Fig 3D, S3 Fig).

To better understand the network-level changes induced by the GAHACR on GA20ox activity we examined the relationships between DEGs to identify transcriptional regulators that could be responsible for the altered GO terms identified using Cytoscape [41] (S4 Fig). We identified a specific cluster of DEGs with related GO-terms 'Response to light' and 'Rhythmic process' that include well-studied components of the circadian clock PSEUDO-RESPONSE REGULATOR 5 (PRR5), PSEUDO-RESPONSE REGULATOR 7 (PRR7), GIGANTEA (GI), and REVEILLE 7 (RVE7). We also observed that two transcriptional regulators PHYTOCHROME INTERACTING FACTOR 4 (PIF4), and PHYTOCHROME RAPIDLY REGULATED1 (PAR1) are upregulated by the GAHACR intervention and could be causal to the changes in transcription

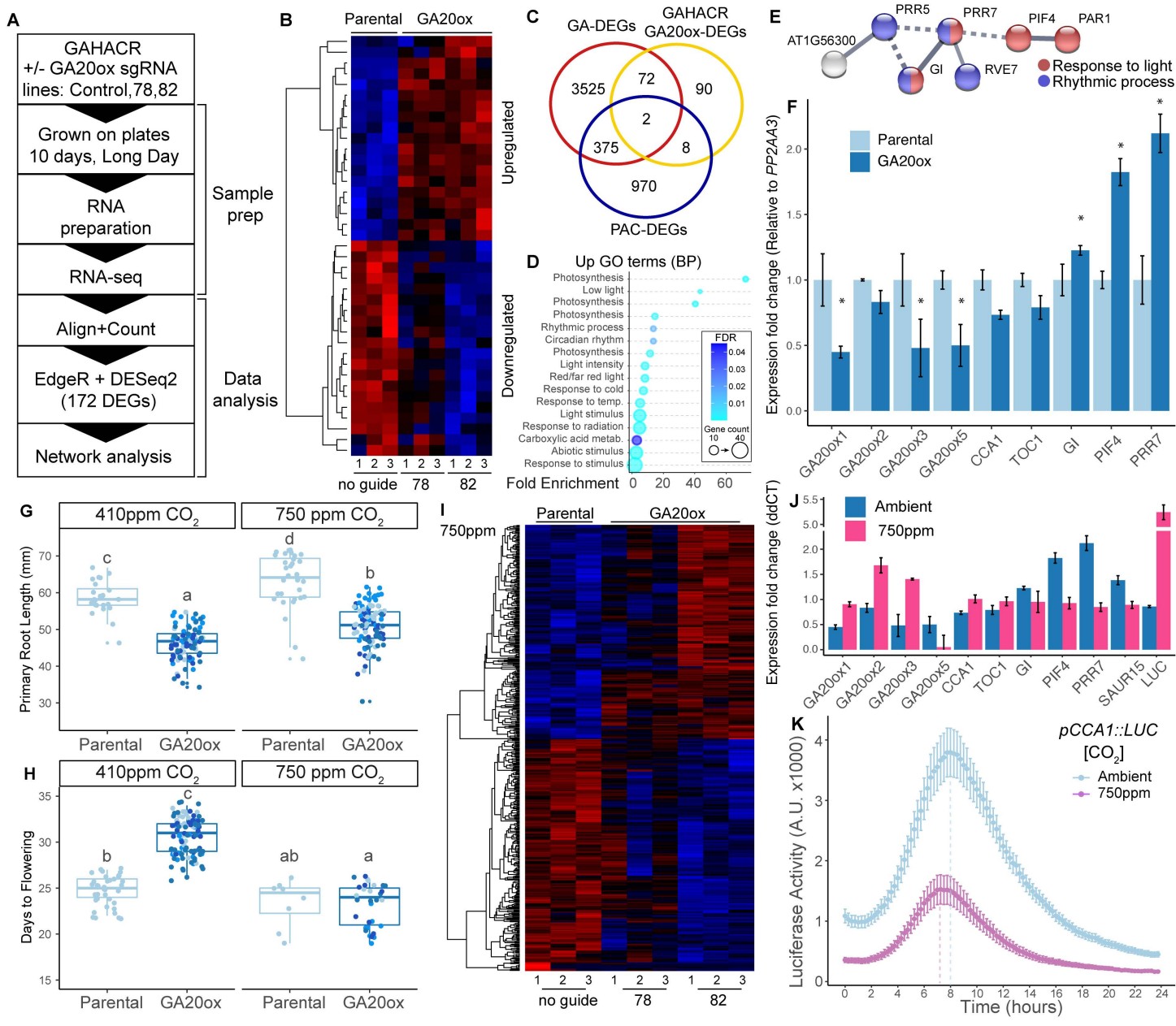

**Fig 3. Elevated carbon dioxide levels suppress the GAHACR modulation of GA20oxidase transcription levels and flowering time. (A)** Schematic description of RNA-seq pipeline, RNA-Seq was analyzed using EdgeR and DESeq2 packages and cross validated. **(B)** Heatmap of top DEGs identified by RNA-seq analysis at ambient $CO_2$ levels. Values are in $Log_2$, where red is upregulated, and blue is downregulated. **(C)** Intersection of GAHACR GA20ox-DEGs with DEGs identified in previously published studies: GA-DEGs [38], PAC-DEGs [39,40]. **(D)** Gene Ontology terms enriched in the upregulated GAHACR targeted to *GA20ox,* FDR – False discovery rate, GO was called via gProfiler. **(E)** Subnetwork of up-regulated GAHACR DEGs enriched in GO terms Response to light (red) and Rhythmic process (blue). **(F)** Differential gene expression analyzed by RT-qPCR in parental GAHACR and GAHACR retargeted to GA20oxidase (line 82) at ambient carbon dioxide levels. Data is plotted as ddCT values, normalized to *AtPP2AA3* transcript levels and then to the parental condition, error bars are standard error. **(G)** Primary root length of 10-day old seedlings and **(H)** days to flowering in plants grown under ambient carbon dioxide (~410 ppm) or elevated carbon dioxide (~750 ppm) levels. Letters indicate statistical significance by ANNOVA. **(I)** Heatmap of top DEGs identified by RNA-seq analysis at 750 ppm carbon dioxide levels. Values are in $Log_2$ where red is upregulated and blue is downregulated. **(J)** Differential gene expression analyzed by RT-qPCR in parental GAHACR and GAHACR retargeted to GA20oxidase (line 82) at elevated carbon dioxide (~750 ppm) levels. Data is plotted as ddCT values, normalized to AtPP2AA3 transcript levels and then to the parental condition, error bars are standard error. **(K)** Time course Luciferase activity analysis of the clock promoter pCCA1 under ambient carbon dioxide (~410 ppm) or elevated

carbon dioxide (~750 ppm) levels. Dotted lines are drawn from the data point with the highest value (peak expression level). Data was collected for three separate runs of the experiment per genotype with 24 biological replicates per experiment, and error bars represent standard error. Vertical lines were graphed according to the highest expression level to demonstrate a subtle shift in peak maximum at high carbon dioxide levels.

related to photosynthesis and light perception. In order to verify these observations, we performed an independent experiment to test RNA levels of selected genes in 10-day old seedlings to examine whether we could replicate these changes in expression by qRT-PCR (Fig 3F). First, we examined the extent to which our GAHACR intervention reduced the expression of the GA20oxidases that they are directly targeting. We observed reductions in RNA abundance by ~50% for *GA20ox1*, *GA20ox3* and *GA20ox5*, with a much weaker reduction in *GA20ox2* abundance (Fig 3F). Next, we examined expression levels of *GI*, *PIF4* and *PRR7* and again saw small but significant increases in transcript abundance (Fig 3F). Because our data pointed to genes in the circadian clock having altered expression levels, we also tested the abundance of core clock genes *CCA1* and *TOC1* and observed that their abundance was slightly lowered (Fig 3F), suggesting that the amplitude or phase of clock components may be slightly altered by this GAHACR intervention.

To understand the connection between elevated atmospheric carbon dioxide and GA-responsive gene expression, we subjected our lines with GAHACR targeted to *GA20oxidase* to growth at elevated carbon dioxide levels. The original experiments demonstrated that elevated carbon dioxide levels rescued seedlings treated with the GA biosynthesis inhibitor paclobutrazol (PAC) in *Arabidopsis* [31,32], which inhibits ent-kaurene oxidase, an enzyme upstream of the GA20oxidase enzyme [11,42]. Therefore, we wanted to test whether elevated carbon dioxide levels would have the same effect on our genetic intervention on GA20oxidase activity. We grew seedlings on plates at elevated carbon dioxide levels (750 ppm) for 10 days and evaluated the effect of primary root growth (Fig 3G). We observed no rescue of primary root growth by elevated carbon dioxide levels at this time point, however we did note that primary root length was increased for both parental and experimental seedlings at elevated carbon dioxide levels (Fig 3G), consistent with previous studies that demonstrate increased biomass and root growth under elevated carbon dioxide levels due to higher photosynthetic rates [43]. However, we did observe a marked rescue of the delay in flowering time under long days at carbon dioxide levels (Fig 3H).

We performed an RNA-seq experiment on 10-day old seedlings grown under elevated carbon dioxide levels and observed differential gene expression specific to the GAHACR intervention (Fig 3I, S2 Table). These DEGs appear to be carbon dioxide level-specific as they are not fully shared with the ambient DEGs (S5, S6A, S6B Figs), suggesting that the molecular phenotype of the GAHACR intervention has been rescued at this stage even through differences in primary root growth have not (Fig 3G). Indeed, in an independent experiment we observed a rescue of *GA20ox1*, *GA20ox2*, and *GA20ox3* mRNA levels, but not *GA20ox5* (Fig 3J). Consistent with this observation we saw a five-fold increase in *LUC* gene expression in lines with GAHACR retargeted to *GA20ox* genes (Fig 3J). Furthermore, this rescue was observed for ambient DEGs *GI*, *PIF4* and *PRR7* as well as the core clock genes *CCA1* and *TOC1* (Fig 3J, S7 Fig). Additionally, an auxin-related up regulated DEG *SAUR15* was also rescued by elevated carbon dioxide levels, suggesting the rescue of GAHACR intervention does not only rely on GA signaling (Fig 3J). When we compared our ambient and elevated carbon dioxide level RNA-seq data sets we observed a large number of differentially expressed genes (>5,000 DEGS, S6C Fig) with many that are circadian regulated using DIURNAL [44], which led us to question whether there was a misalignment in timing between these datasets, or whether carbon dioxide level could influence either the phase or amplitude of the circadian clock. To test this hypothesis we employed a transcriptional reporter of the circadian clock the *pCCA1::LUC* under ambient and elevated carbon dioxide levels [45]. Surprisingly, under long day entrainment we observed a dampening of the amplitude of the *pCCA1::LUC* reporter under elevated carbon dioxide levels (Fig 3K, light purple), and a slightly earlier max peak in phase (~30 minutes, difference in dashed lines) compared to ambient levels (Fig 3K, light blue). Taken together with the differences in RNA-seq data sets as well as independent RT-qPCR analyses, we suggest that

atmospheric carbon dioxide levels may slightly influence the circadian clock, and that the GA feedback pathway appears to be downstream of this modulation.

## Discussion

In this study, we implemented a rewiring of the endogenous GA feedback network using the GA-regulated Hormone Activated CAS9-based Repressor (GAHACR) described in our previous study [34]. To test the importance of transcriptional feedback in the GA signaling pathway, we implemented a mathematical model of the GA phytohormone pathway to best predict the impact of targeting a GA-regulated Hormone Activated CAS9-based Repressor (GAHACR) to different nodes in the GA biosynthesis pathway (Fig 1). Based upon these predictions, we built and implemented the perturbation predicted by the model to lower GA levels *in planta*, by targeting *GA20 oxidase* (*GA20ox*) and *GID1* for repression by GAHACRs. We observed alterations in primary root length, flowering time and global gene expression changes in a manner consistent with GA-feedback reprogramming that validates this forward engineering approach (Fig 2). Interestingly, we observed that reducing the positive feedback between DELLAs and *GA20ox* transcription uncovered a connection between GA and the circadian clock that can be rescued by elevated $CO_2$ concentration (Fig 3). The use of HACRs here builds out our ability to use synthetic signaling engineering solutions to engineer morphology in a multicellular organism in a model-driven manner.

Cells use a diverse set of feedback control mechanisms that are often layered in multiple architectures and scales to produce robustness in the face of environmental and biotic heterogeneity [46,47]. Feedback loops in transcriptional systems are fundamentally required to detect and reset any deviation from set-point operation [46], and the *Arabidopsis* GA biosynthesis and signaling pathway exhibits both negative and positive feedback within the pathway (DELLAs negative regulate their own transcription, and positively regulate biosynthesis genes, [24]). Feedback regulation is required to generate signal robustness across input conditions [47], yet probing these networks without modifying node, or gene, activity is currently impossible within endemic systems. We have employed the GAHACR as a novel node in the GA network that effectively reduces the gain in the positive regulation step of the feedback system (Fig 1). This leaves the endogenous node and network intact, which is a feature of dCAS9-based transcription factors that we have previously employed in re-engineering the auxin phytohormone signaling pathway [34]. In the GA pathway, we have mathematically and experimentally demonstrated that the gain in feedback from the signaling pathway is required for GA signal output levels, consistent with current models [24].

Elevated carbon dioxide levels reverse low GA phenotypes caused by biosynthesis mutants in tomato [33], and *Arabidopsis* plants treated with the GA biosynthesis inhibitor Paclobutrazol (PAC, [31,32]). Here, we demonstrate that feedback modulation by the GAHACR on *GA20ox* family genes can also be reversed at elevated carbon dioxide levels. Intriguingly, this rescue is observable in flowering time, but not earlier in plant development such as in root development (Fig 3G, 3H). This suggests that lines of feedback within the GA signaling pathway are also sensitive to carbon dioxide levels, in part through PIF4, which was also previously observed [31]. Surprisingly, we observed components of the circadian clock regulation machinery (*PRR5, PRR7, GI, RVE7*) were altered in plants with *GA20ox* feedback modulation. Several studies have shown a link between the clock and GA reviewed in [48]. For example GA signaling oscillates due to regulation of the GA receptors [26], and the DELLA protein REPRESSOR OF ga1–3 (RGA) interacts with CCA1 [49], but GA has not been observed to directly modulate to the core clock output. Instead, GA is hypothesized to feedback to clock outputs by regulating the oscillatory expression of clock-regulated genes [26]. We suspect that the cluster of genes identified through this line of feedback (*PRR5, PRR7, GI, RVE7*) may represent genes of this class, and the elevated levels of *BBX18*, a known clock modulator [50], may provide a clue to the altered network behavior at high carbon dioxide levels (Fig 3, S6C Fig). These results collectively point to multiple independent lines of $CO_2$ signaling inputs and new opportunities for chronoculture engineering [51].

The Green Revolution leveraged diverse mutations in the GA pathway across multiple species [8–10], raising the question of how robustly these modifications in crop species will perform under changing conditions during climate change, particularly

elevated levels of atmospheric carbon dioxide. Current evidence suggests that the canonical Green Revolution GA pathway dwarfing alleles employed in monocots – such as *sd1* in rice and *Rht-B1b/D1b* in wheat – have not been systematically evaluated for altered performance under elevated $CO_2$. While numerous FACE (Free-Air $CO_2$ Enrichment) and chamber studies in rice, wheat, and barley document cultivar-specific responses to rising $CO_2$, these experiments do not explicitly identify or genotype the GA-related alleles present in their tested cultivars [52–54]. As a result, although phenotypic trends suggest that $CO_2$ enrichment may modify height, lodging, and biomass allocation – traits historically shaped by GA pathway mutations – direct evidence that elevated $CO_2$ is eroding, amplifying, or otherwise modifying the benefits conferred by monocot Green Revolution dwarfing alleles must be directly tested. Climate models predict an increase in atmospheric carbon dioxide to reach 600–900 ppm by the year 2100 unless drastic measures are taken to curb emissions (IPPC, AR6 [55]), and indeed ambient carbon dioxide levels have risen appreciably (2017 ~ 408 ppm vs. 2024 ~ 425 ppm) since the outset of this study. While physiological effects of elevated carbon dioxide levels on plants are starting to come into focus [56,57], the modulation of GA pathway mutations are likely to be failure prone under elevation past evolved set points based upon our results and previous studies in eudicots [32,33]). We hope that this study raises testable hypotheses for GA network engineering of both eudicot and monocot crop species for desired agronomic traits, i.e., [58]. The gene network identified here (Fig 3E) are gene candidates that could be targeted in engineering and selective breeding efforts to potentially minimize these deleterious effects in crops plants.

## Methods

### Model construction

To build our models we first implemented the differential equations described in the Middleton et al. in a python environment for ease of use. To simulate the GAHACR, we incorporated additional equations that were identical to those used to simulate the transcription, translation, and GA induced degradation of DELLA. A version of the model that set the GAHACR protein and mRNA concentration to zero was used to simulate wildtype regulation. Additionally, a version of the model where the GA dependent degradation terms associated with the GAHACR were set to zero was used to simulate the No Degron CR. To simulate the GAHACR's or No Degron CR's repression of GA20ox expression, we incorporated a term that captures GAHACR protein concentration scaled by a user defined repression strength constant into the denominator of the hill function used to simulate the activation of GA20ox by the DELLA protein [59]. The same constants used in the original Middleton et. al model was preserved in our model. Those associated with the GAHACR either mimicked those of the DELLA proteins or were programmatically varied across a range as reported in Fig 1C. Additional terms are described in the supplemental materials under Supplemental Model Information.

### Construction of plasmids

Expression cassettes for the gRNAs were cloned by golden gate methodology as described in [60]. The construction of the GAHACR was described previously [34]. The sgRNA expression cassettes contain sgRNAs driven by the U6 promoter and have a U6 terminator and were cloned as described in [60], with the modification that the dCAS9 cassette was eliminated from the final T-DNA by mutagenesis PCR. The guide RNA sequences in the GA20ox sgRNA plasmid are as follows: guide1: GA20ox5 – TATGATTTTGAAAACAAACG, guide2: GA20ox3 – AATTTATTTAGTGGCTGAAC, guide3: GA20ox1 – ATGGTCCTTTTAGTCTTTAT. The guide RNA sequences in the GID1 sgRNA plasmid are as follows: guide1: GID1A – GCGTTGAGGGATGAGTAGGG, guide2: GID1B – AAGAAAGACCAATCGGACGG, guide3: GID1C – GTGGATAAGAATATCGGCGT.

### Construction of plant lines

All GAHACR reporter lines were built by transforming the yeast artificial chromosome plasmids described above into Agrobacterium tumefaciens (GV3101) and using the resulting strains to transform a Columbia-0 background by floral dip [61]

as described in [34]. Transformants carrying guide RNA constructs were then selected using a light pulse selection [62]. Briefly, this involves exposing the seeds to light for 6 hours after stratification (4°C for 2 days in the dark) followed by a three-day dark treatment. Resistant seedlings demonstrate hypocotyl elongation on Hygromycin. After selection seedlings were transplanted to soil and grown in long day conditions at 22°C. Two different auxin HACR backgrounds (independent transformant lines) were transformed with a sgRNA targeting GA20ox or GID1.

## Characterizing plant phenotypes

To characterize root growth phenotypes in plant lines with and without a GAHACR regulating target genes, we selected T2 transformants for lines that had a gRNA targeting GA pathway genes and the parental HACR background that had no gRNA. These seedlings were grown on selection plates for 4 days before being transplanted to media lacking selection. In all cases the parental controls that lack a gRNA and the lines derived from them were all grown in parallel and phenotyped on the same day to ensure the data collected was comparable. For flowering time experiments, T2 plants were grown on selection media for 7 days before being transplanted to soil and grown until flowering in a Conviron growth chamber with either long or short daylight settings. All plants that were phenotyped in soil were grown on Sunshine #4 mix soil in rose pots and watered every other day and checked every day for the presence of the inflorescence bolt at greater than 1 cm above the rosette, at which point they were scored as flowering. For RNA-seq experiments on seedlings grown at elevated carbon dioxide, T4 plants were grown on media without selection for 4 days to ensure proper germination before being transplanted to new plates lacking selection and grown in a Conviron growth chamber with either ambient or elevated carbon dioxide settings and long day light settings.

## qPCR

All qPCR assays were performed on seedlings grown in their stated conditions and then flash frozen in liquid nitrogen. Seedlings were homogenized in 2 mL tubes using a mixer mill bead beater with one steel ball bearing, at max speed for two 3-minute beating sessions, separated by a bath in liquid nitrogen to maintain freezing conditions. RNA was extracted from these seedlings using the Illustra RNAspin Mini Kit from GE (Via Millipore/Sigma Aldrich, Cat. No. GE25-0500-71). cDNA was then prepared from 1 µg of RNA using the iScript cDNA synthesis kit (Biorad, USA) and then used to run a qPCR with the iQ SYBR Green Supermix (Biorad, USA) on a qPCR machine (Biorad, USA). Each sample was analyzed for expression of target genes and AtPP2AA3 subunit which was used to normalize mRNA levels. A standard curve was generated using the pooled cDNA samples for each primer set to determine amplification efficiency and only primers with >90% efficiency were used. The primers used are listed in the supplementary primers table.

## RNA-Seq

For ambient carbon dioxide concentrations, T4 seedlings were cultured on plates for 4 days without drug selection, and then transplanted to LS0 plates without drug for 10 days in either ambient or carbon dioxide supplemented Conviron (Pembina, North Dakota) growth chambers. RNA was extracted using the Illustra RNAspin Mini Kit from GE, and RNA-Seq was performed by Amaryllis Nucleics (Oakland CA). In brief, RNA was checked for quality and quantity on a Bioanalyzer, and poly(A) mRNA was purified from total RNA. The library was constructed using the Amaryllis proprietary construction kit, followed by QC by E-gel & Bioanalyzer. Libraries were pooled and run on an Illumina NextSeq 500 SR75, with a target of approximately 20 million reads per sample. Read preprocessing, mapping and differential gene expression was performed by Amaryllis, in addition to independent differential gene expression analysis in house using EdgeR and DESeq2. Network analysis was performed in cytoscape. GO terminology enrichment was calculated using g:profiler (biit. cs.ut.ee/gprofiler/gost) and plotted in R using ggplot2 (see code deposited at https://github.com/achillobator/GAHACR).

## Luciferase assays

Luciferase based time course assays were used to characterize the dynamics of the CCA1 promoter driving luciferase (*CCA1::LUC*) under ambient and elevated carbon dioxide concentrations. *CCA1::LUC* seedlings were grown in sterile 24 well plates in 0.5x LS0 media in either ambient or 750 ppm carbon dioxide levels for 10 days in long day conditions (6h-22h in light). These were then sprayed with luciferin (5 μM in water, Biosynth, United Kingdom, Cat. No. L-8220) in the evening of the final day (approximately 7 hours before midnight) and loaded into a Tecan (Spark model, Tecan Life Sciences, Switzerland), where they were sampled over time for 24 hours starting at time zero (midnight). The acquisition settings were as follows: Kinetic mode, Luminescence (Luciferin1), Interval time – 15 minutes, settle time – 0, Integration time – 1000 milliseconds, Output – counts/s. Data was collected for three separate runs of the experiment per genotype with 24 biological replicates per experiment. Data was analyzed in R, using the ggplot2 package (see code deposited at https://github.com/achillobator/GAHACR).

## Plant genotype list

| Plant genotype | Used in the following Fig |
| --- | --- |
| GAHACR (GAI1) parental: PHD3 (p2301Y-tOCS-pUBQ1:NLS-Venus-LucPlus-tUBQ1-pU6:pUBQ1_gRNA_Target1-tU6-pUBQ10:dCas9-GAI1-TPLRD2-tNos) | Fig 2C–2E |
| GAHACR (GAI1) + GA20ox sgRNAs: PHD3 (p2301Y-tOCS-pUBQ1:NLS-Venus-LucPlus-tUBQ1-pU6:pUBQ1_gRNA_Target1-tU6-pUBQ10:dCas9-GAI1-TPLRD2-tNos), plus GA20oxidase sgRNAs (pHEE401_pU6_sgRNAGA20ox1,2,3,5) | Fig 2C–2E |
| GAHACR (GAI1) + GID1 sgRNAs: PHD3 (p2301Y-tOCS-pUBQ1:NLS-Venus-LucPlus-tUBQ1-pU6:pUBQ1_gRNA_Target1-tU6-pUBQ10:dCas9-GAI1-TPLRD2-tNos), plus GID1 sgRNAs (pHEE401_pU6_sgRNAGID1A/B/C) | Fig 2C–2E |
| GAHACR (RGA1) parental: PHD6 (p2301Y-tOCS-pUBQ1:NLS-Venus-LucPlus-tUBQ1-pU6:pUBQ1_gRNA_Target1-tU6-pUBQ10:dCas9-RGA1-TPLRD2-tNos) | Fig 2C–2J, Fig 3B, Fig 3F–3J |
| GAHACR (RGA1) + GA20ox sgRNAs: PHD3 (p2301Y-tOCS-pUBQ1:NLS-Venus-LucPlus-tUBQ1-pU6:pUBQ1_gRNA_Target1-tU6-pUBQ10:dCas9-GAI1-TPLRD2-tNos), plus GA20oxidase sgRNAs (pHEE401_pU6_sgRNAGA20ox1,2,3,5) | Fig 2C–2J, Fig 3B, Fig 3F–3J |
| GAHACR (RGA1) + GID1 sgRNAs: PHD3 (p2301Y-tOCS-pUBQ1:NLS-Venus-LucPlus-tUBQ1-pU6:pUBQ1_gRNA_Target1-tU6-pUBQ10:dCas9-GAI1-TPLRD2-tNos), plus GID1 sgRNAs (pHEE401_pU6_sgRNAGID1A/B/C) | Fig 2C–2E |
| *pCCA1::LUC* | Fig 3K |

## Plasmid maps

pHEE401 modified to eliminate pEC1:dCAS9 (pNL1897) - https://benchling.com/s/seq-IE5V76ktpYAi2Vo1LgRE?m=slm-MDrDwUciEEBK1S7CSr61

GA20ox Guide RNA plasmid - https://benchling.com/s/seq-Chse2EBNCleOFNBEvx3p?m=slm-J1ja71DHmewEB1YWxFpe

GID Guide RNA plasmid - https://benchling.com/s/seq-jFMbFK58Bg6RPIinSCxp?m=slm-fDt6B5K5njnTEFGRp5Rk

## Supporting information

**S1 Fig. Schematic of GA-HACR retargeting to endogenous genes in *Arabidopsis*.** The top schematic of the genetic circuit used to build GA responsive HACRs (above dotted line) was described in [34]. In the lower portion of the schematic (below the dotted line) demonstrates the additional pAt-U6 driven gRNA which targets the endogenous genes (i.e., GA20ox or GID1). This allows the GA-HACR to simultaneously act not only as a reporter via Venus/Luciferase, but also to modify existing genetic networks.
(TIF)

**S2 Fig. RNA-Sequencing analysis of GAHACR network at ambient carbon dioxide levels – Identifying differentially expressed genes. A-B.** Selected DEGS were chosen to be graphed to demonstrate gene expression changes across the two selected lines (78 and 82) and wild type. We selected 10 down-regulated **(A)** and 10 up-regulated **(B)** DEGs, and graphed the counts per million across the 3 replicates in standard boxplots to demonstrate the similar trends across the two lines. **C.** To ensure that we were robustly detecting the maximum impact of the GA-HACR intervention, we applied two supplemental DEG finding packages, DESeq2 and EdgeR, and plotted the overlap in detected DEGs. **D.** A more detailed breakdown of the DEGs identified in the two DEG caller methods separated by genetic line.
(TIF)

**S3 Fig. RNA-Sequencing analysis of GAHACR network at ambient carbon dioxide levels – Gene ontology analysis.** Gene Ontology terms enriched in the upregulated (A) and downregulated (B) differentially expressed genes from GAHACR targeted to *GA20ox* at ambient carbon dioxide levels. FDR – False discovery rate, GO was called via gProfiler. Data was graphed in R, using the ggplots2 package (see methods).
(TIF)

**S4 Fig. Network analysis of the GAHACR network at ambient carbon dioxide levels.** DEGs from the GA-HACR RNA-sequencing performed at ambient carbon dioxide levels were imported into Cytoscape for network analysis using the STRING database. All singletons were trimmed, and functional enrichment was performed. A subnetwork of Rhythmic processes and Light responsive proteins were identified and excerpted into Fig 3E. Nodes are color coded based on the log fold change observed in the GA-HACR lines 82 versus the parental GA-HACR line. DEGs that demonstrated a reduction in expression are colored blue and DEGs that increased expression are red (see scale).
(TIF)

**S5 Fig. RNA-Sequencing analysis of GAHACR network at elevated carbon dioxide levels – Identifying differentially expressed genes. A-B.** Selected DEGS were chosen to be graphed to demonstrate gene expression changes across the two selected lines (78 and 82) and wild type. We selected 10 down-regulated **(A)** and 10 up-regulated **(B)** DEGs and graphed the counts per million across the 3 replicates in standard boxplots to demonstrate the similar trends across the two lines. **C.** A more detailed breakdown of the DEGs identified in the two genetic lines at elevated carbon dioxide demonstrates the overlap in DEGs lists. **D.** The DEGS lists from the ambient and elevated (750 ppm) carbon dioxide treatments were compared to determine if there is significant overlap between these datasets.
(TIF)

**S6 Fig. Analysis of the GAHACR network at elevated carbon dioxide levels.** DEGs from the GA-HACR RNA-sequencing performed at elevated (750 ppm $CO_2$) carbon dioxide levels were imported into Cytoscape for network analysis using the STRING database. All singletons were trimmed, and functional enrichment was performed. The network was clustered using MCL (inflation value = 4). A subnetwork of Rhythmic processes was identified (bottom right, biological rhythms) that includes the genes PRR5 and RVE1. Nodes are color coded based on the log fold change observed in the GA-HACR lines 82 versus the parental GA-HACR line. DEGs that demonstrated a reduction in expression are colored blue and DEGs that increased expression are red (see scale). **B**. Heatmap of top upregulated DEGs identified by RNA-seq analysis at ambient $CO_2$ levels (left 6 columns). Only line 82 upregulated DEGs are shown for simplicity, and many appear to be reduced in expression at elevated carbon dioxide levels (right 6 columns). Values are in Log2, where red is upregulated, and blue is downregulated. **C.** Intersection of DEGs generated by comparing ambient to elevated carbon dioxide demonstrate a large number of genes that are differentially expressed regardless of the GAHACR intervention.
(TIF)

**S7 Fig. Analysis of the GAHACR combined network at both carbon dioxide levels.** DEGs from the GA-HACR RNA-sequencing performed at ambient and elevated (750 ppm $CO_2$) carbon dioxide levels were pooled and imported into Cytoscape for network analysis using the STRING database. All singletons were trimmed, and functional enrichment was performed. The network was clustered using MCL (inflation value = 4), and we manually subset selected clusters to examine how elevated carbon dioxide influences the networks. Each cluster node was colored based on the differential gene expression from the ambient (left column) or elevated (right column) no guide versus line 82 GAHACR experiments. **A.** Cluster defined by GO term Cellular response to hypoxia, GO:0071456, FDR = 1.02E-9. **B.** Cluster defined by GO Cellular component keyword Photosystem, GO:0009521, FDR = 3.12E-20 **C.** Cluster defined by UniProt keyword Biological Rhythms, KW-0090, FDR = 8.27E-16.
(TIF)

**S1 Table. Datafile of DEGs for RNA-Sequencing analysis of GAHACR network at ambient carbon dioxide levels.**
(XLSX)

**S2 Table. Datafile of DEGs for RNA-Sequencing analysis of GAHACR network at elevated carbon dioxide levels.**
(XLSX)

**S3 Table. Primer Table.** Oligonucleotides used in this study.
(PDF)

**S8 Model Information. Gibberellin model.** Gibberellin model from Middleton et. al. [24] with new terms utilized in this study.
(PDF)

## Author contributions

**Conceptualization:** Alexander R. Leydon, Arjun Khakhar, Jennifer L. Nemhauser.

**Data curation:** Alexander R. Leydon, Leonel Flores.

**Formal analysis:** Alexander R. Leydon, Leonel Flores, Arjun Khakhar.

**Funding acquisition:** Jennifer L. Nemhauser.

**Investigation:** Alexander R. Leydon, Leonel Flores, Arjun Khakhar.

**Methodology:** Alexander R. Leydon.

**Project administration:** Alexander R. Leydon, Jennifer L. Nemhauser.

**Resources:** Jennifer L. Nemhauser.

**Software:** Alexander R. Leydon, Arjun Khakhar.

**Supervision:** Alexander R. Leydon, Jennifer L. Nemhauser.

**Validation:** Alexander R. Leydon.

**Visualization:** Alexander R. Leydon, Leonel Flores, Arjun Khakhar.

**Writing – original draft:** Alexander R. Leydon, Leonel Flores, Arjun Khakhar.

**Writing – review & editing:** Alexander R. Leydon, Jennifer L. Nemhauser.

## Acknowledgments

We thank Dr. Takato Imazumi for sharing *pCCA1*::*LUC* reporter lines and advice, particularly on luminescence measurements; Dr. Adam Steinbrenner for discussions on figures and results. We would like to thank Andrew Lemmex for their

contribution in building initial plant strains. We would like to thank Tajinder Ubhi for GO graph inspiration and discussion of figures.

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
