## [Decision Letter · Decision Letter 0]

4 Sep 2025

Dear Dr. Nemhauser,

Thank you for submitting your manuscript to PLOS ONE. After careful consideration, we feel that it has merit but does not fully meet PLOS ONE’s publication criteria as it currently stands. Therefore, we invite you to submit a revised version of the manuscript that addresses the points raised during the review process.

We look forward to receiving your revised manuscript.

Kind regards,

Md. Mahmudul Hasan, PhD

Academic Editor

PLOS ONE

Journal Requirements:

3. Thank you for stating the following financial disclosure: [This work was supported by grants to JLN from the National Science Foundation (MCB- 1411949) and the National Institutes of Health (R01-GM107084, and R35-GM148135-01), as well as support from the Howard Hughes Medical Institute Faculty Scholars

Program to JLN. ARL was supported as a Simons Foundation Fellow of the Life Sciences Research Foundation]. 

4. Thank you for stating the following in the Acknowledgments Section of your manuscript: [This work was supported by grants to JLN from the National Science Foundation (MCB[1]1411949) and the National Institutes of Health (R01-GM107084, and R35-GM148135- 01), as well as support from the Howard Hughes Medical Institute Faculty Scholars Program to JLN. ARL was supported as a Simons Foundation Fellow of the Life Sciences Research Foundation]

Please remove any funding-related text from the manuscript and let us know how you would like to update your Funding Statement. Currently, your Funding Statement reads as follows: [This work was supported by grants to JLN from the National Science Foundation (MCB- 1411949) and the National Institutes of Health (R01-GM107084, and R35-GM148135-01), as well as support from the Howard Hughes Medical Institute Faculty Scholars

Program to JLN. ARL was supported as a Simons Foundation Fellow of the Life Sciences Research Foundation]. 

5. We notice that your supplementary figures and tables are included in the manuscript file. Please remove them and upload them with the file type 'Supporting Information'. Please ensure that each Supporting Information file has a legend listed in the manuscript after the references list.

6. We note that there is identifying data in the Supporting Information file <20250205_Supplemental Materials_GAHACR.docx>. Due to the inclusion of these potentially identifying data, we have removed this file from your file inventory. Prior to sharing human research participant data, authors should consult with an ethics committee to ensure data are shared in accordance with participant consent and all applicable local laws.

-Location data

Please remove or anonymize all personal information (Name) ensure that the data shared are in accordance with participant consent, and re-upload a fully anonymized data set. Please note that spreadsheet columns with personal information must be removed and not hidden as all hidden columns will appear in the published file.

Reviewers' comments:

Reviewer's Responses to Questions

**Comments to the Author**

1. Is the manuscript technically sound, and do the data support the conclusions?

Reviewer #1: Yes

Reviewer #2: Yes

2. Has the statistical analysis been performed appropriately and rigorously?

Reviewer #1: Yes

Reviewer #2: No

3. Have the authors made all data underlying the findings in their manuscript fully available?

Reviewer #1: Yes

Reviewer #2: Yes

4. Is the manuscript presented in an intelligible fashion and written in standard English?

Reviewer #1: Yes

Reviewer #2: Yes

Reviewer #1: The manuscript of Nemhauser and co-authors is a systems biology study focused on understanding/manipulating gibberellin (GA) homeostasis in Arabidopsis through bioengineering. The authors (previously) constructed a GA-susceptible repressor (GAHACR) and now use it to interfere with GA signalling, correlating results with the fenotypic output. Most important is that the study is mathematical model-driven. The text is well-written, results are statistically sound, and conclusions are based on the experimental data and are fair and convincing. The work and the manuscript are perfectly suitable for the PLOS ONE scope and can be published as an article.

I suggest only minor improvements that would improve the manuscript's readability and expand the audience covered.

First, it would be better to use nomenclature/abbreviations consistently, italicise the gene and species names, remove capitalisations, place space uniformly before "ppm", and use a subscript with CO2 and Log2. Please change the acknowledgement to Dr. Imazumi, specifying "luminescence measurements" instead of "luciferase imaging".

It would be beneficial to display the GA chemical structure, e.g., in Fig.1, and merge panels D and E into a single panel. Additionally, the scheme in Fig. 2 Supp 1 is more visually appealing than Fig. 1B. In panel E, please replace "Without GAHACR" with "Wild type".

Please comment (Fig.3G and 3H and text in the results/discussion) not only on the "Days to Flowering" impact of the elevated CO2, but also how the CO2 level impacts the primary root length - it seems to be promoted.

Although it is referenced and available on github, the model would be better to describe in more details (formula, not the code) in the Methods and/or supplementary section.

Finally, a comment is advisable on comparing GA-metabolism reprogramming and potential CO2 level impact not only in eudicots, but also in monocots. Are they not expected to be different? If so, Arabidopsis/eudicots specification should be added to the manuscript title.

Reviewer #2: Dear authors, you mention an ANOVA and present a heat map; however, I believe that, for the development of this work, more robust statistical analyses could be applied, which would enhance the discussion. Examples include RDA or machine learning approaches such as random forest or LDA, in order to make better use of the data. The statistical analysis you presented appears rather simplistic.

**Do you want your identity to be public for this peer review?** For information about this choice, including consent withdrawal, please see our Privacy Policy

Reviewer #1: No

Reviewer #2: **Yes: ** Deborah Bambil

---

## [Author Response · Author response to Decision Letter 1]

12 Oct 2025

Response to reviews

Reviewer #1 (R1): 

The manuscript of Nemhauser and co-authors is a systems biology study focused on understanding/manipulating gibberellin (GA) homeostasis in Arabidopsis through bioengineering. The authors (previously) constructed a GA-susceptible repressor (GAHACR) and now use it to interfere with GA signalling, correlating results with the phenotypic output. Most important is that the study is mathematical model-driven. The text is well-written, results are statistically sound, and conclusions are based on the experimental data and are fair and convincing. The work and the manuscript are perfectly suitable for the PLOS ONE scope and can be published as an article.

I suggest only minor improvements that would improve the manuscript's readability and expand the audience covered.

Our response: We thank the reviewer for their support.

R1: First, it would be better to use nomenclature/abbreviations consistently, italicise the gene and species names, remove capitalisations, place space uniformly before "ppm", and use a subscript with CO2 and Log2. Please change the acknowledgement to Dr. Imazumi, specifying "luminescence measurements" instead of "luciferase imaging".

Our response: We have made all of the suggested changes to the manuscript. Specifically, regarding nomenclature we use GAHACR in capitalized form to match the usage in our previous publication (eLife 2018). Following standard Arabidopsis nomenclature, genes are capitalized and italicized (e.g., GAI1), proteins are capitalized and not italicized (e.g., GAI1). We have endeavored to ensure this is true throughout the manuscript.

R1: It would be beneficial to display the GA chemical structure, e.g., in Fig.1, and merge panels D and E into a single panel. Additionally, the scheme in Fig. 2 Supp 1 is more visually appealing than Fig. 1B. In panel E, please replace "Without GAHACR" with "Wild type".

Our response: Thank you for your suggestions. We have included the GA4 structure in Fig. 1B. As our paper does not go into depth on the chemistry of the GA biosynthetic pathway, we hope that our inclusion of several in-depth reviews of the chemistry is sufficient for interested readers. We have adapted the text and the panel order as requested. For simplicity and compactness of the figure, we have left the cartoons of the repressors as they are.

R1: Please comment (Fig.3G and 3H and text in the results/discussion) not only on the "Days to Flowering" impact of the elevated CO2, but also how the CO2 level impacts the primary root length - it seems to be promoted.

Our response: We agree that we see an increased root length under supplemental carbon dioxide applications, and this has been seen in previous studies of carbon dioxide seeding experiments - reviewed well in 10.1042/BCJ20220245. We have commented on this in the results section and the discussion, added this citation, and hope that readers will be interested in why the flowering time is rescued independently from the root growth phenotypes.

R1: Although it is referenced and available on github, the model would be better to describe in more details (formula, not the code) in the Methods and/or supplementary section.

Our response: We have provided the original model, descriptions of terms, and our new term in the supplemental material.

R1: Finally, a comment is advisable on comparing GA-metabolism reprogramming and potential CO2 level impact not only in eudicots, but also in monocots. Are they not expected to be different? If so, Arabidopsis/eudicots specification should be added to the manuscript title.

Our response: The reviewer brings up a very important point, that in our opinion has not been directly investigated in the literature. It is unclear if CO2 levels will alter monocots in the same way as has been documented for dicots, and whether a similar feedback network will exist. We added additional language to the discussion to directly address this concern, which we hope monocot specialists will take note of, and therefore test directly in their systems, especially for the well described GA modulated crops in wheat and rice (such as sd1 in rice and Rht-B1b/D1b in wheat). As the title is already quite long, we have added the species name into the abstract, so that readers will know the study was done in Arabidopsis (eudicot).

--

Reviewer #2: Dear authors, you mention an ANOVA and present a heat map; however, I believe that, for the development of this work, more robust statistical analyses could be applied, which would enhance the discussion. Examples include RDA or machine learning approaches such as random forest or LDA, in order to make better use of the data. The statistical analysis you presented appears rather simplistic.

Our response: We understand that there is confusion about the statistical methods used to generate the data in figure 3. The only place where data was analyzed by ANOVA is in the statistical comparison of seedlings grown with and without supplemental carbon dioxide, and is only mentioned in the figure legend with respect to panels G & H. The RNA seq data was analyzed using DESeq2 (DOI: 10.18129/B9.bioc.DESeq2) and EdgeR (DOI: 10.18129/B9.bioc.edgeR) packages, which are considered standard in the field, and is diagrammed in the flowchart in figure 3A. We have added additional text to clarify how all data were analyzed.

---

## [Decision Letter · Decision Letter 1]

10 Nov 2025

Reprogramming feedback strength in gibberellin biosynthesis highlights conditional regulation by the circadian clock and carbon dioxide.

PONE-D-25-38292R1

DearDr. Jennifer Nemhauser,

We’re pleased to inform you that your manuscript has been judged scientifically suitable for publication and will be formally accepted for publication once it meets all outstanding technical requirements.

Kind regards,

Md. Mahmudul Hasan, PhD

Academic Editor

PLOS ONE

Additional Editor Comments (optional):

Reviewers' comments:

Reviewer's Responses to Questions

**Comments to the Author**

Reviewer #1: All comments have been addressed

2. Is the manuscript technically sound, and do the data support the conclusions?

Reviewer #1: Yes

3. Has the statistical analysis been performed appropriately and rigorously?

Reviewer #1: Yes

4. Have the authors made all data underlying the findings in their manuscript fully available?

Reviewer #1: Yes

5. Is the manuscript presented in an intelligible fashion and written in standard English?

Reviewer #1: Yes

Reviewer #1: I am satisfied the way the authors responded to my review. The paper can be accepted the way it is now. My congratulations to the authors!

**Do you want your identity to be public for this peer review?** For information about this choice, including consent withdrawal, please see our Privacy Policy

Reviewer #1: No

---

## [Editor Report · Acceptance letter]

PONE-D-25-38292R1

PLOS ONE

Dear Dr. Nemhauser,

I'm pleased to inform you that your manuscript has been deemed suitable for publication in PLOS ONE. Congratulations! Your manuscript is now being handed over to our production team.

Kind regards,

on behalf of

Dr. Md. Mahmudul Hasan

Academic Editor

PLOS ONE